# A Simple and Effective Bioassay Method Suitable to Comparative In Vitro Study of Tomato Salt Tolerance at Early Development Stages

**DOI:** 10.3390/mps5010011

**Published:** 2022-01-19

**Authors:** Marat R. Khaliluev, Liliya R. Bogoutdinova, Galina N. Raldugina, Ekaterina N. Baranova

**Affiliations:** 1Laboratory of Plant Cell Engineering, All-Russia Research Institute of Agricultural Biotechnology, Timiryazevskaya 42, 127550 Moscow, Russia; 2Department of Biotechnology, Institute of Agrobiotechnology, Russian State Agrarian University—Moscow Timiryazev Agricultural Academy, Timiryazevskaya 49, 127550 Moscow, Russia; 3Laboratory of Plant Cell Biology, All-Russia Research Institute of Agricultural Biotechnology, Timiryazevskaya 42, 127550 Moscow, Russia; bogoutdinova_lr@rambler.ru (L.R.B.); greenpro2007@rambler.ru (E.N.B.); 4Laboratory of Ion Transport and Salinity Resistance, K. A. Timiryazev Institute of Plant Physiology, Russian Academy of Sciences, Botanicheskaya Street 35, 127276 Moscow, Russia; raldugina42@mail.ru; 5Laboratory of Plant Protection, N.V. Tsitsin Main Botanical Garden of Russian Academy of Sciences, Botanicheskaya 4, 127276 Moscow, Russia

**Keywords:** *Solanum lycopersicum* L., NaCl stress in vitro, respiratory and photosynthetic CO_2_ gas exchange, proline, photosynthetic pigments, ultrastructure of cotyledon spongy mesophyll

## Abstract

In vitro evaluation of tomato seeds and seedlings for salt tolerance has undoubted advantages (high productivity, as well as stability and reproducibility of the obtained experimental data due to the maintenance of constant controlled conditions) in comparison with open-field system and pot experiments. However, even high-quality seeds greatly differ in the uniformity of germination capacity and germination energy. Heterogeneous germination in the habit and developmental stage of plant material significantly distorts the obtaining of relevant experimental data suitable for correct interpretation. In our study, we propose a simple and effective bioassay method suitable to comparative in vitro study of tomato salt tolerance using shoot apex of seedlings at the early first-true-leaf stage. Shoot apexes cultured the on the root induction medium (RIM) supplemented with 0.2 mg/L indole-3-butyric acid (IBA) and NaCl at different concentrations (0–250 mM NaCl) revealed significant differences between two tomato genotypes (line YaLF and cv. Rekordsmen) at the organismal (measurements of CO_2_ gas exchange), organ (rhizogenesis frequency; number and length of de novo regenerated roots; root fresh (RFW) and dry (RDW) weights; shoot fresh (SFW) and dry (SDW) weights), tissue (the average cross-sectional area of epidermal and mesophylls cotyledonary cells) and cellular (ultrastructure of chloroplast and nuclear compartments) development levels. In addition, a quantitative comparison of proline and photosynthetic pigments contents under 75 and 150 mm NaCl treatments showed a different response between two tomato genotypes. The proposed methodological approach can be used for other plants with a high response to auxin-induced rhizogenesis in vitro, as well as for the comparative in vitro assessment of other abiotic stresses.

## 1. Introduction

Globally, tomatoes (*Solanum lycopersicum* L.) are the second most important vegetable crops after potatoes and are used as a valuable source of vitamins and lycopene in fresh or processed tomato products [1,2]. In 2019, the world’s total tomato fruit production was approximately 180.7 million tons, which accounts for 1.7% (3.0 million tons) from Russia. Russia ranks 11th in the world for tomatoes harvested area (81.3 thousand ha) [3]. Tomatoes are predominantly grown in the soil for open-field system (80.4% from total tomato fruit production) in the south regions of the country (Krasnodar and Stavropol territories, as well as Volga and the central black earth regions), the lands of which are more or less subject to primary and/or secondary soil salinization.

In addition to its great practical value, the tomato has been widely used as a model plant in various basic research investigated the underlying the mechanisms of plant resistance to abiotic stresses, including salinity [4,5]. This is due to the large number of morphological traits clearly identifiable at different ontogenetic stages [6,7], detailed molecular genetic maps [8,9], and reproducible in vitro tissue culture techniques [10,11].

The tomato (*S. lycopersicum* L.) is rated as a high salt sensitive crop. Plant growth inhibition, leading to significant reduction in tomato yield, occurs when the electrical conductivity (ECe) threshold of 2.5 dS/m [12], that corresponds to low salinity according to the FAO (USDA) soil classification (ECe = 2.0–4.0 dS/m) [13].

An important component underlying the mechanisms of plant resistance to salinity is comprehensive assessment based on biochemical, physiological, anatomical, genetic, and molecular responses. Creating and maintaining controlled conditions are an essential requirement to obtain reliability experimental data suitable for correct interpretation. This is due to the fact that environmental changes (temperature, relative humidity, light intensity, level of air pollution, etc.) dramatically affect a plant’s response to salinity [14,15]. In vitro testing of plant genotypes under stress conditions devoid of all of the mentioned above disadvantages. Additionally, a comparative assessment for salinity tolerance, as well as screening of highly salinity-resistant genotypes, can be conducted already at the early development stages (seeds and seedlings), which are the most critical for tomatoes [16,17,18]. It is also important that a clear correlation between tomato salt tolerance under in vitro (callus) and in vivo (plants grown in greenhouse) conditions has been observed [18,19,20]. 

Many studies carried out to testing of tomato salinity tolerance during germination of non-sterile seeds under stress treatment [21,22,23] or aseptically seed germinated on culture media, containing various NaCl or Na_2_SO_4_ concentrations [24,25,26,27]. However, even high-quality seeds greatly differ in the uniformity of germination capacity and germination energy. Even greater differences in these parameters are observed after surface seed sterilization and in vitro germination. Plant material that is heterogeneous in habit and developmental stage significantly distorts the obtaining of relevant experimental data suitable for correct interpretation. In our study, we propose a simple and effective bioassay method suitable to comparative in vitro study of tomato salt tolerance using shoot apex of seedlings at the early first-true-leaf stage. This short-term bioassay method allows researchers to obtain correct experimental data for morphometric, physiological, biochemical and cytological analyzes at the cellular, tissue, organ, and organismal development levels. 

## 2. Materials and Methods

### 2.1. Plant Material and Obtaining of Aseptic Donor Seedlings

The seeds of tomato (*S. lycopersicum* L.) line YaLF, the male parental line for the commercial F1 Yunior hybrid, and cv. Rekordsmen were obtained from N.N. Timofeev breeding station, Russian State Agrarian University—Moscow Timiryazev Agricultural Academy (Moscow, Russia), as well as All-Russia Research Institute of Irrigated Vegetable, Melon and Ground Growing (Astrakhan oblast, Kamyziyak, Russia), respectively. F1 Yunior hybrid, and cv. Rekordsmen are recommended for cultivation in the greenhouse and open-field system, respectively. In vitro aseptic donor seedlings were produced by surface sterilization of tomato seeds in 96% ethanol for 30 s and in 20% water solution (*v*/*v*) of a commercial bleach Ace (5% NaOCl, Procter and Gamble, Saint Petersburg, Russia) supplemented with a 5 µL of Tween-20 for 6–8 min. The sterilized seeds were washed with distilled water four times for 1 min each and then germinated in culture vessels containing agar-solidified (0.7% (*w*/*v*)) MS basal medium [28] without plant growth regulators (PGR). The pH was adjusted to 5.7–5.8 before autoclaving at 121 °C for 20 min. The cultures were maintained in a climate chamber WLR-351H (Sanyo, Japan) under 25/23 (day/night) ±1 °C, with fluorescence light (65 µmol m^−2^s^−1^) during long-day photoperiod (16 h light/8 h dark).

### 2.2. Effects of NaCl Treatments on De Novo Root Formation in In Vitro Tomato Seedlings and Its Morphological Characteristics

Roots and part of the hypocotyl were excised from 8–10-day-old aseptic tomato seedlings at the early first-true-leaf stage, after which shoot fragments 1.5–2 cm in length were transferred into culture vessels (300 cm^3^) containing root induction medium (RIM) (MS medium with half strength of macro and micro salts, vitamins, 2% (*w*/*v*) sucrose, 0.7% (*w*/*v*) agar, 0.2 mg/L indole-3-butyric acid (IBA) (Sigma, Saint-Louis, MO, USA) and 25–300 mM NaCl) (Figure 1). RIM without NaCl was used as a control. IBA solution were dissolved in distilled water, filter-sterilized with MCE membrane (0.22 μm Millipore, Burlington, IA, USA) and stored until use at −20 °C. 

After 8 d of in vitro culture, the evaluated morphological characteristics included the rhizogenesis frequency (%), number and length (cm) of regenerated roots, root fresh (RFW) and dry (RDW) weights, as well as shoot fresh (SFW) and dry (SDW) weights. The rhizogenesis frequency (%) was determined as the ratio between the number of seedlings with root formation and the total number of seedlings. In addition, time of the beginning root formation was noted. The FW and DW were determined gravimetrically using an analytical balance (Sartorius, Göttingen, Germany). To determine the DW, roots and shoots were dried at 65 °C until a constant weight. Each variant of treatment (*n* = 10) was performed in three replications.

### 2.3. Measurements of CO_2_ Gas Exchange

Measurements of respiratory and photosynthetic CO_2_ gas exchange in in vitro tomato seedlings was carried out using earlier developed whole-plant chamber system [29,30]. This closed chamber system includes the following components connected in series: sealed culture vessels, pump, air dryer, rotameter, GOA-4 infrared gas analyzer (Khimavtomatika, Russia) with a 0–0.05% CO_2_ scale, two-position gas switch for multiple measurements. The annular fluorescent lamp LUMILUX T9 L 32W/840 C G10Q (OSRAM, Germany) was applied for photosynthetic CO_2_ gas exchange measurements. Tomato seedlings of the studied genotypes after 8 d culture on RIM without NaCl, as well as with the addition of experimentally established sublethal (150 and 250 mM NaCl for the line YaLF and cv. Rekordsmen, respectively) and intermediate NaCl concentrations were used for assessment. 24 h before measurements the culture vessels with tomato seedlings were unsealed and kept open under growth chamber. Seedlings were incubated in complete darkness 2 h before measurements. CO_2_ was recorded under dark and light conditions at a constant temperature (22–23 °C) for 5 min. Dark respiration rates (DRR) (CO_2_ gas exchange (μg/h) in the darkness) and true photosynthetic rates (TPR) (CO_2_ gas exchange (μg/h) under light and dark conditions) was determined per mg of seedling dry weight. Each variant of treatment (*n* = 10) was performed in three replications.

### 2.4. Proline and Photosynthetic Pigment Contents

Leaves of tomato seedlings cultured on the RIM supplemented with 0, 75, and 150 mM NaCl were used for biochemical assays. 

Determination of free proline content was carried out with a ninhydrin-based protocol [31] with some modifications. Proline extraction was carried out by boiling a 200 mg leafy sample in 4 mL distilled water, and after cooling a ninhydrin reagent (1.25 g ninhydrin, 20 mL 6M H_3_PO_4_, 30 mL glacial acetic acid) was added. The color intensity was determined by Specol-11 spectrophotometer (Carl Zeiss, Oberkochen, Germany) at a wavelength of 520 nm against a sample in which distilled water was added instead of the extract. The proline content (µM g^−1^ of FW) was determined from a calibration curve using proline (Serva, Heidelberg, Germany).

The photosynthetic pigment contents (chlorophylls a (Chla), b (Chlb)and carotenoids (Car)) were determined by extracting pigments from leaves with 96% ethyl alcohol [32]. The degree of solution absorption (optical density) for chlorophylls a, b, and carotenoids was determined using Genesys 20 spectrophotometer (ThermoScientific, Waltham, MA, USA) at a wavelength of 665, 649 and 471 nm, respectively. The pigment content (µg g^−1^ FW) was calculated by the formulas [33]: Cchl a = 13.70D_665_ − 5.76 D_649_;
Cchl b = 25.80 D_649_ − 7.60 D_665_;
Ccar = (1000D_471_×2.13Cchl a − 97.64Cchl b)/209
A = C ∗ V/1000 ∗ n
where C, pigment concentrations; D, optical density; V, extract volume; and n, leaf fresh weight.

### 2.5. Preparation of Cotyledon Samples for Light and Transmission Electron Microscopy (TEM)

Excised samples from the middle part of the cotyledons (2–3 mm) were taken from seedlings after eight days culture on RIM including 0 (control), 75, and 150 mM NaCl and fixed for 24 h in 2.5% glutaraldehyde (Merck, Darmstadt, Germany) dissolved in 0.1 M Sorensen’s phosphate buffer (pH 7.2) with 1.5% sucrose. Then the samples were washed, post-fixed in 1% OsO4 (Sigma-Aldrich, Saint-Louis, MO, USA), and dehydrated in ethanol of increasing concentrations (30, 50, 70, 96, and 100%) and in propylene oxide (Fluka, Darmstadt, Germany). The samples were embedded in Epon-812 and Araldite 502 mixture (Merck, Darmstadt, Germany) according to the standard procedure. For light microscopy, semi-thin sections (1–2 μm) were prepared using glass knives and ultramicrotome LKB-V (LKB, Bromma, Sweden), placed on glass slides and embedded in epoxy resin. Samples were photographed using Olympus BX51 microscope (Olympus, Tokyo, Japan) equipped with Color View II camera (Soft Imaging System, Münster, Germany). The average cross-sectional area of upper epidermis (UE), spongy (SM) and palisade (PM) mesophylls was determined using a Cell A software package (Olympus, Japan). At least 300 cells of mentioned above tissues from three independent seedlings for each experimental treatment were analyzed.

For electron microscopy, embedded samples were sectioned with ULTRA 45° diamond knife (Diatom, Nidau, Switzerland), using LKB-V ultramicrotome (LKB, Bromma, Sweden), placed on formvar coated blends and stained with uranyl acetate and lead citrate [34]. Thin sections were analyzed and photographed with H-500 electron microscope (Hitachi, Tokyo, Japan) at accelerating potential of 75 kV.

### 2.6. Statistical Treatments of Experimental Data

Statistical treatments of experimental data were performed at 5% significance level using the analysis of variance (ANOVA) and Duncan’s multiple range tests with AGROS software (version 2.11, Moscow, Russia), as well as standard MS Excel software packages.

## 3. Results

### 3.1. Influence of Different NaCl Concentrations on the Number of Regenerated Roots and Their Morphological Characteristics

NaCl concentrations that inhibit in vitro root organogenesis in the studied tomato genotypes during cultivation of seedlings on the RIM were determined. Thus, necrosis of the hypocotyl fragment in direct contact with the culture medium was observed on the 5th day of culture in the seedling of line YaLF under 200 mM NaCl exposure, which led to almost complete inhibition of root formation. A slight decrease of rhizogenesis frequency to 93.3% was demonstrated in fragments of tomato seedlings cv. Rekordsmen on RIM with a higher NaCl content (250 mM), compared with YaLF line. High concentrations of NaCl (150 mM and more) not only reduced the rhizogenesis frequency, but also lengthened the timing of root formation (Table 1).

Dramatic genotypic differences by the number of regenerated roots and their length were revealed (Figure 2). The lowest concentration of NaCl in RIM (25 mM) led to a significant reduce the number of regenerated roots in seedlings YaLF line as compared to the control. At the same time, a significant increase in their length was observed. A subsequent reduction of in a root number in the tomato YaLF line was noted under 150 mM NaCl treatment (Figure 2a). Compared to the controls, reduction in a root number of the cv. Rekordsmen seedlings occurred only under 150 mM NaCl exposure, and the minimal NaCl concentration did not lead to a significant increase in their length (Figure 2b). The formation of shortened roots in both tomato genotypes was revealed during the culture of seedlings on RIM supplemented with NaCl at concentrations of higher than 75 mM. 

Salinity treatments induced by NaCl had a significant effect on the root fresh weight (RFW) for two tomato genotypes (Table 2). 

An inverse relationship was found between RFW and intensity of NaCl salinity. At the same time, no significant differences were found between the studied tomato genotypes. On the contrary, dramatic genotypic differences by root dry weight (RDW) were found. In general, both tomato genotypes were characterized by a decrease in RDW with increasing NaCl concentration in the RIM. However, compared with the control, a significant decrease of RDW has already mentioned in culturing seedlings of the YaLF line on RIM supplemented with the lowest NaCl concentration (25 mM), whereas for the cv. Rekordsmen only under 150 mM NaCl salinity exposure.

### 3.2. Influence of Different NaCl Concentrations on the Shoot Fresh Weight (SFW) and Shoot Dry Weight (SDW)

The results of two-way ANOVA test showed statistical differences at 5% significance level in SFW and SDW between both the studied tomato genotypes and the NaCl concentrations in RIM. In addition, the differences were significant for interaction «genotype × culture medium». Generally, SFW of both tomato genotypes reduced with increasing of NaCl concentration in the RIM (Figure 3a). Differences between tomato genotypes were assessed in the NaCl concentration, at which there was a significant decrease in SFW compared to the control. Thus, these values were 50 and 75 mM NaCl for the line YaLF and cv. Recordsmen, respectively. In addition, the SFW of cv. Rekordsmen was significantly higher than that of the line YaLF when the seedlings cultured on RIM containing NaCl at a concentration of 50 mM and higher, with the exception of 100 mM NaCl exposure. 

The dramatic difference between tomato genotypes was observed by a change in the SDW under salt treatments (Figure 3b). As in the case of the SFW, reduce of SDW in the line YaLF occurred under weak salinity (50 mM NaCl). On the contrary, there were no statistically significant differences between the control and experimental treatments for SDW in the cv. Rekordsmen. It should be noted that seedlings of both tomato genotypes cultured on the RIM containing 100 mM NaCl and higher had cotyledon with obvious signs of chlorosis, the extent of which becomes more pronounced under increased stressful salinity conditions. Moreover, the growth of true leaves was inhibited in tomato seedlings under moderate salinity (100–150 mM NaCl), while their formation did not occur by 200–250 mM NaCl treatments.

Accordingly, based on the mentioned above of experimental data sets on rhizogenesis frequency, as well as morphological characteristics of the tomato roots and shoots, sublethal concentrations 150 and 250 mM NaCl were revealed for the YaLF line and cv. Rekordsmen, respectively. 

### 3.3. Influence of NaCl Salinity on the DRR and TPR

Respiratory and photosynthetic CO_2_ gas exchange in in vitro tomato seedlings was assessed using intermediate and sublethal NaCl concentrations experimentally established for each genotype (Figure 4). Significant reduction of TPR and DRR (by 1.1 and 1.3 times, respectively) was observed during cultivation of tomato seedlings line YaLF under 75 mM NaCl treatment compared to control. A subsequent decrease of TPR in seedlings line YaLF was noted on a RIM containing a sublethal NaCl concentration (Figure 4a). Compared with the line YaLF, significant differences in TPR and DRR for tomato seedlings of the cv. Rekordsmen between control conditions, as well as 75 and 150 mM NaCl treatments was not found. The change of respiratory and photosynthetic CO_2_ gas exchange in this genotype occurred only under 250 mM NaCl (Figure 4b).

### 3.4. Influence of NaCl Salinity on the Proline and Photosynthetic Pigment Contents

The dramatic differences between tomato genotypes were observed by a change in the contents of proline and Chla under salt treatments (Figure 5a,b). Significant reduction of proline (by 1.4 and 3.1 times) and Chla (by 1.9 and 1.6 times) contents occurred during cultured of tomato seedlings line YaLF under 75 and 150 mM NaCl treatment compared to control. On the contrary, the content of proline in leaves of tomato cv. Rekordsmen significantly increased under 75 mM NaCl, while the content of Chla under 75 and 150 mM NaCl treatments did not differ from the control values. 

Salinity did not change the Chlb content (Figure 5c) and significantly increased Car concentrations (Figure 5d) in tomato leaves of both genotypes. 

### 3.5. The Morphological Response of Epidermal and Mesophylls Cotyledonary Tomato Cells to NaCl Salinity In Vitro

Generally, histological analysis revealed that the epidermal and mesophylls cotyledonary cells in plants of cv. Rekordsmen were less sensitive to presence of NaCl in the RIM, compared with line YaLF (Figure 6). Thus, cotyledonary UE cells in tomato line YaLF were characterized by gradual decrease the average cross-sectional area under NaCl salinity. Thus, the average cross-sectional area of UE under 75 and 150 mM NaCl were significantly less (1.2 and 1.4 times, respectively) compared to control conditions. A similar response of cotyledonary UE cells to NaCl treatments was observed in the tomato cv. Rekordsmen. However, 150 mM NaCl treatment resulted in a statistically increase in the cell size of this tissue compared with line YaLF. 

The dramatic differences between tomato genotypes in response to NaCl of cotyledonary SM and PM cells were also established. Shape and size (a decrease the average cell cross-sectional area by almost 2.6 and 2.7 times as compared to the control conditions) changes of the PM in line YaLF occurred under 75 and 150 mM NaCl. An almost two-fold decrease in the average cross-sectional area of SM was also caused by these NaCl concentrations. Size of PM and SM cells in cotyledon leaves of cv. Rekordsmen under 75 mM NaCl impact was unchanged. Compared with the line YaLF, 150 mM NaCl salinity caused approximately 1.5-fold increase in the average cross-sectional area of cotyledonary PM cells in tomato plants cv. Rekordsmen.

### 3.6. Ultrastructure of Cotyledonary SM Cells from Control and Salt-Treated (150 mM NaCl) Tomato Seedlings

The TEM images demonstrate the comparative structural organization of chloroplast and nuclear compartments in the cotyledonary SM cells of control and salt-treated (150 mM NaCl) tomato seedlings of the line YaLF (Figure 7) and cv. Rekordsmen (Figure 8). Under control conditions, chloroplasts located in a thin layer of cytoplasm between the cell wall and a large central vacuole have lenticular shape typical for this tissue. In chloroplasts, distinguishable thylakoid grana stacks and located between them stromal thylakoids, starch grains, plastoglobules, as well as small electron-lucent regions in the stroma containing nucleoids are clearly visible (Figure 7a and Figure 8a). The nucleus has a typical structure in the cotyledonary SM cells: a small granular nucleolus, a thin layer of tightly condensed chromatin organized in the nuclear periphery and associated with the nuclear membrane, and a nucleoplasm filled with dispersed and less packed euchromatin (Figure 7b and Figure 8b). 

Salinity-induced irreversible ultrastructural changes of some SM cells in line YaLF were established (Figure 7c,e). Figure 7f shows a part of a dead cell in which cytoplasm is completely absent. At the edge of the dead cells, swollen and rounded chloroplast is observed. Some SM cells showed clearly visible signs of convex plasmolysis, such as noticeable cytoplasmic invaginations, plasma membrane dissociation with the cell wall, and highly compacted chloroplasts with electron-dense thylakoids, which still maintain the plasma membrane integrity (Figure 7f). The noted ultrastructural disturbances indicate that 150 mM NaCl salinity leads to significant cell damage in the photosynthetic tissues of cotyledons line YaLF. These changes affect not only the structural and functional disturbances of the chloroplast and nuclear compartments, as well as the inhibition of biosynthetic processes in them (and also the death of a number of cells).

Chloroplasts in cotyledonary SM cells of the cv. Rekordsmen did not change in shape and structure under 150 NaCl treatment compared with the line YaLF. The differences were found only in a decrease of size plastoglobuli (Figure 8c). The nuclear compartment also retained its characteristic shape and location under salinity. However, as in the case of YaLF line, structural changes in condensed chromatin (formation of large compacted lumps) occurred. The nucleolus contains fibrillar component with the lack of a granular component, which indicates disorders of ribosome biogenesis (Figure 8d). Thus, in comparison with the YaLF line, disorganization of the chloroplast and nuclear structure, leading to their destruction, does not occur in the cotyledonary SM cells of the cv. Rekordsmen under 150 mM NaCl treatment. The ultrastructural changes are due to metabolic disturbance caused by exposure to salt stress.

Thus, based on a comparative assessment, tomato cv. Rekordsmen characterized by enhanced resistance to NaCl salinity compared to the YaLF line on the organismal (TPR and DRR of seedlings), organ (rhizogenesis frequency; number and length of de novo regenerated roots; root fresh (RFW) and dry (RDW) weights; shoot fresh (SFW) and dry (SDW) weights), tissue (the average cross-sectional area of epidermal and mesophylls cotyledonary cells), and cellular (ultrastructure of chloroplast and nuclear compartments) development levels.

## 4. Discussion

Existing methods for assessment of tomato salt tolerance are divided into direct (tomato yield and productivity as well as biometric characteristics of plant growth and biomass; determination of seed germination under saline conditions) and indirect (physiological, biochemical, cytological and other characteristics that correlate with direct assessment indicators) assessments [35,36]. At the same time, comparative testing of tomato genotypes is carried out in the open-field system [37,38,39], as well as in pot experiments under greenhouse [40,41], hydroponic [42,43,44,45,46], or natural environmental conditions [47] (Figure 9). In addition, preliminary laboratory experiments are widely used to evaluate the salt tolerance of germinating tomato seeds or seedlings. 

In vitro evaluation of tomato genotypes for salt tolerance has undoubted advantages (high productivity, as well as stability and reproducibility of the obtained experimental data due to the maintenance of constant controlled conditions) in comparison with open-field system and pot experiments (Figure 9). The authors cultivated seeds [21,22,23], shoot apexes [48], shoot apical meristem [49], and callus [50,51,52,53] as an explant source under salinity treatments in vitro. The disadvantage of seeds as an explant source is high heterogeneity, as well as greatly differ in the uniformity of germination capacity and germination energy. Callus tissue cells are also characterized by high heterogeneity. However, a number of studies have shown clear correlation between tomato salt tolerance under in vitro and in vivo conditions [18,19,20]. Heterogeneous plant material in habit and developmental stage significantly distorts of relevant experimental data suitable for correct interpretation under salinity in vitro. For a comparative in vitro study of tomato salt tolerance, we used shoot apex 1.5–2 cm in length of seedlings at the early first-true-leaf stage. Aseptic donor seedlings of two tomato genotypes (line YaLF and cv. Rekordsmen) of different ecological and geographical origin (Central and Volgo-Vyatka regions of Russia, respectively) were used. The soil salinity levels of these regions differ significantly, as a result of which the studied genotypes may differ significantly in salinity tolerance. Thus, more than 31% of the soils in the Astrakhan region are characterized by a high salinity (the concentration of sodium and sulfate ions reaches up to 7.1 and 12.5 mM per 100 g of soil, respectively), as well as about 20% of solonetzic soil complexes [54]. This assumption was used as the basis for the choice of plant material. 

The proposed methodological bioassay made it possible to reveal significant differences between tomato genotypes for salt tolerance using various morphological, physiological, biochemical and cytological characteristics. Cultivation of the shoot apexes on the RIM supplemented with various NaCl concentrations revealed significant differences between tomato genotypes at the whole organism (TPR and DRR of seedlings), organ (rhizogenesis frequency; number and length of de novo regenerated roots; root fresh (RFW) and dry (RDW) weights; shoot fresh (SFW) and dry (SDW) weights), tissue (the average cross-sectional area of epidermal and mesophylls cotyledonary cells) and cellular (ultrastructure of chloroplast and nuclear compartments) development levels. In addition, a quantitative comparison of proline and photosynthetic pigments contents under 75 and 150 mm NaCl treatments showed a different response between tomato genotypes. Determined rhizogenesis-inhibiting NaCl concentrations and morphometric characteristics of regenerated roots under NaCl-salinity suggest that root growth is the most indicative parameter for evaluating tomato salt tolerance in vitro [47,53]. The presented effective bioassay method was previously tested for evaluating some qualitative and quantitative cytological characterization of tomato roots (cv. Rekordsmen) de novo regenerated under 25–250 mM NaCl salinity [55], as well as for comparative anatomical and morphological studies of the epidermal and cortical parenchyma hypocotyl cells of tomato line YaLF and cv. Rekordsmen [56].

## 5. Conclusions

Therefore, we have developed and tested a simple and effective bioassay method suitable to comparative in vitro study of tomato salt tolerance using shoot apex of seedlings at the early first-true-leaf stage. This short-term bioassay method allows researchers to obtain correct experimental data for morphometric, physiological, biochemical, and cytological analyses at the different organization levels. The proposed methodological approach can be used for other plants with a high response to auxin-induced rhizogenesis in vitro. In addition, it can also be useful for the comparative in vitro assessment of other abiotic stresses, such as exposure to heavy metals or PEG-induced osmotic stress, as well as for a comparative assessment of control and transgenic plants expressing heterologous genes.

## Figures and Tables

**Figure 1 mps-05-00011-f001:**
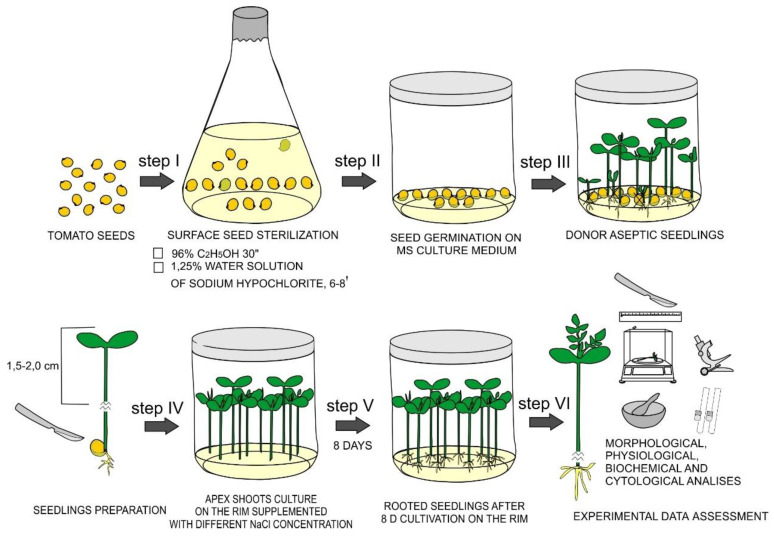
Experimental design for comparative in vitro assessment of tomato salt tolerance at early development stages.

**Figure 2 mps-05-00011-f002:**
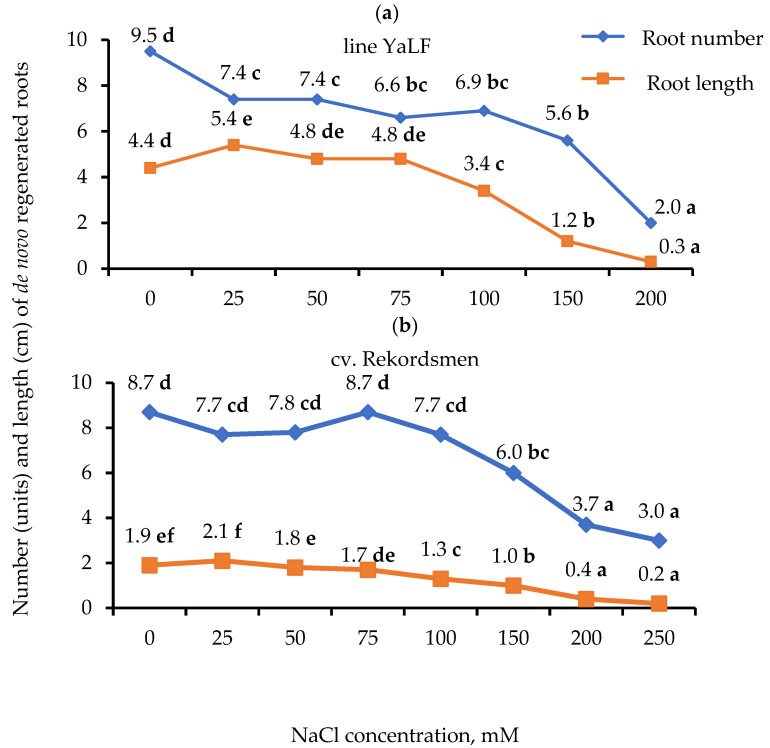
Influence of different NaCl concentrations in RIM on the number (units) and length (cm) of de novo regenerated roots in seedlings of the YaLF line (**a**) and cv. Rekordsmen (**b**). Means followed by the same letter are not significantly different at α = 0.05 according to the Duncan’s multiple range test (*n* = 30).

**Figure 3 mps-05-00011-f003:**
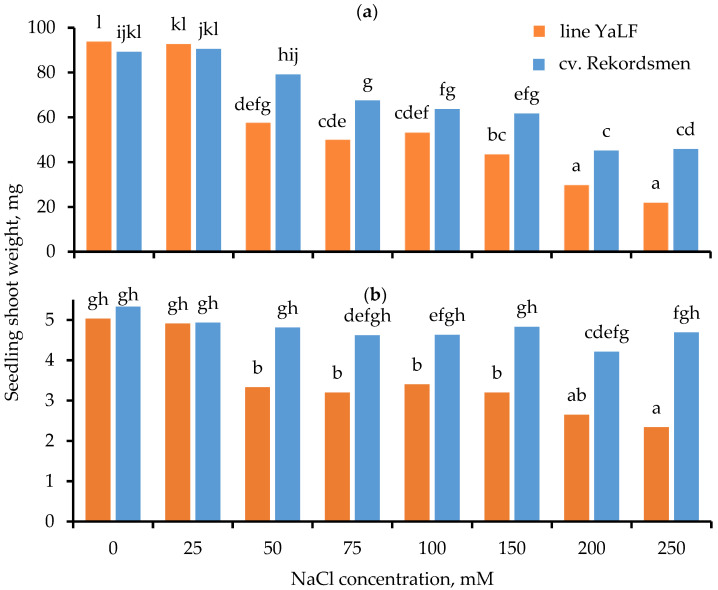
Influence of different NaCl concentrations in RIM on the SFW (**a**) and SDW (**b**) of tomato seedlings line YaLF and cv. Rekordsmen. Means followed by the same letter are not significantly different at α = 0.05 according to the Duncan’s multiple range test (*n* = 30).

**Figure 4 mps-05-00011-f004:**
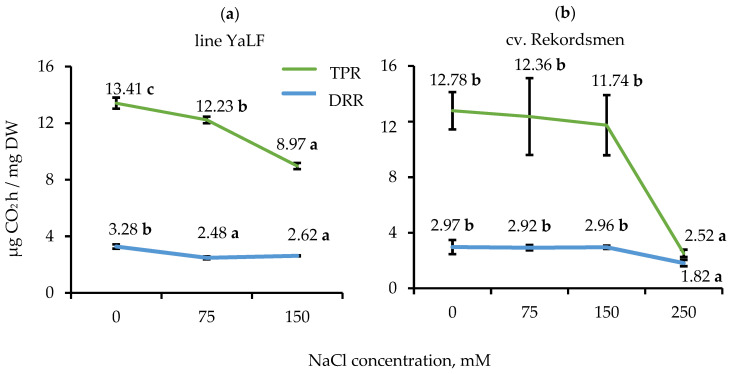
Influence of different NaCl concentrations in RIM on the TPR and DRR of tomato seedlings line YaLF (**a**) and cv. Rekordsmen (**b**). Means ± standard errors followed by the same letter are not significantly different at α = 0.05 according to the Duncan’s multiple range test (*n* = 30).

**Figure 5 mps-05-00011-f005:**
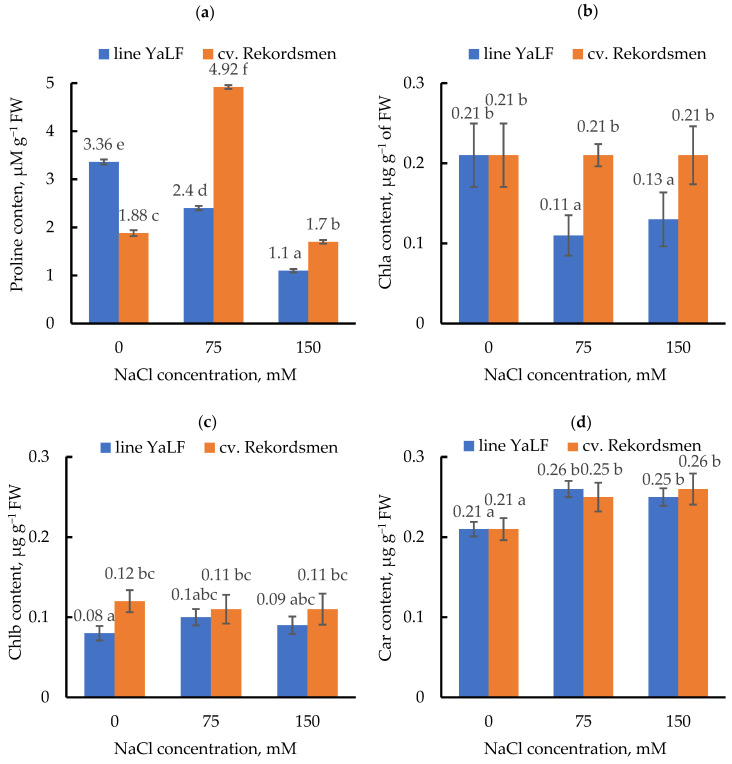
Influence of NaCl salinity on the content of proline (**a**), Chla (**b**) and Chlb (**c**), as well as Car (**d**) from tomato leaves. Means ± standard errors at α = 0.05 according to ANOVA tests are presented.

**Figure 6 mps-05-00011-f006:**
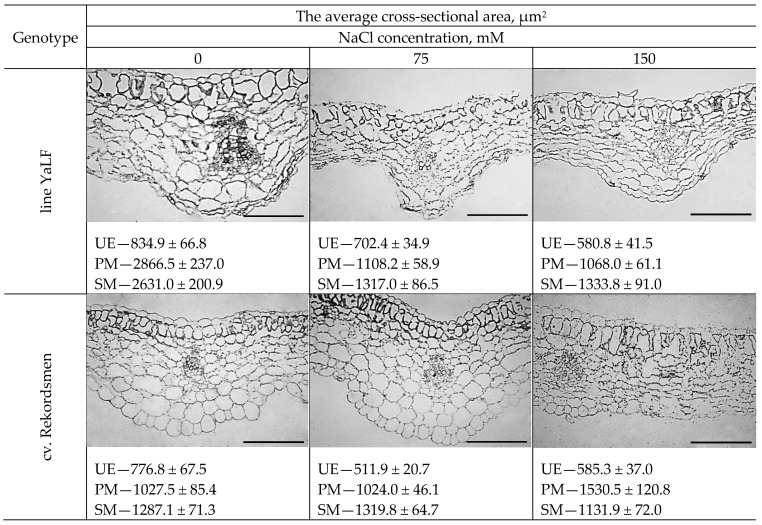
Comparison of a cotyledonous leaf cross-section at middle part from tomato seedlings line YaLF and cv. Rekordsmen after 8 d cultivation on the RIM supplemented with various NaCl (0, 75 and 150 mM) concentrations. UE, upper epidermis; PM, palisade mesophyll; SM, spongy mesophyll. Means of the average cross-sectional area ± standard errors at α = 0.05 according to ANOVA tests are presented (*n* ≥ 300). Scale—100 µm.

**Figure 7 mps-05-00011-f007:**
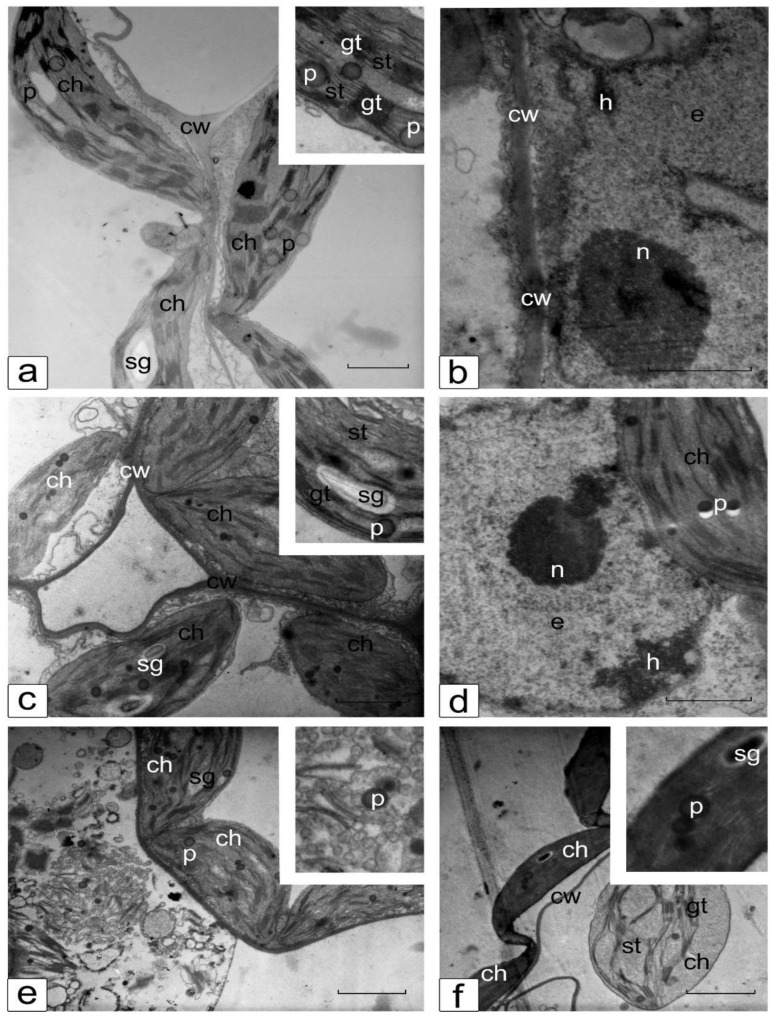
Ultrastructure of chloroplast (**a**,**c**,**e**,**f**) and nuclear (**b**,**d**) compartments in the cotyledonary SM cells of control (**a**,**b**) and salt-treated (150 mM NaCl) (**c**–**f**) tomato seedlings of the line YaLF. Ch, chloroplast; gt, st, granal and stromal thylakoids, respectively; cw, cell wall; e, euchromatin; h, heterochromatin; n, nucleolus; p, plastoglobuli; sg, starch grain. Scale, 1 µm.

**Figure 8 mps-05-00011-f008:**
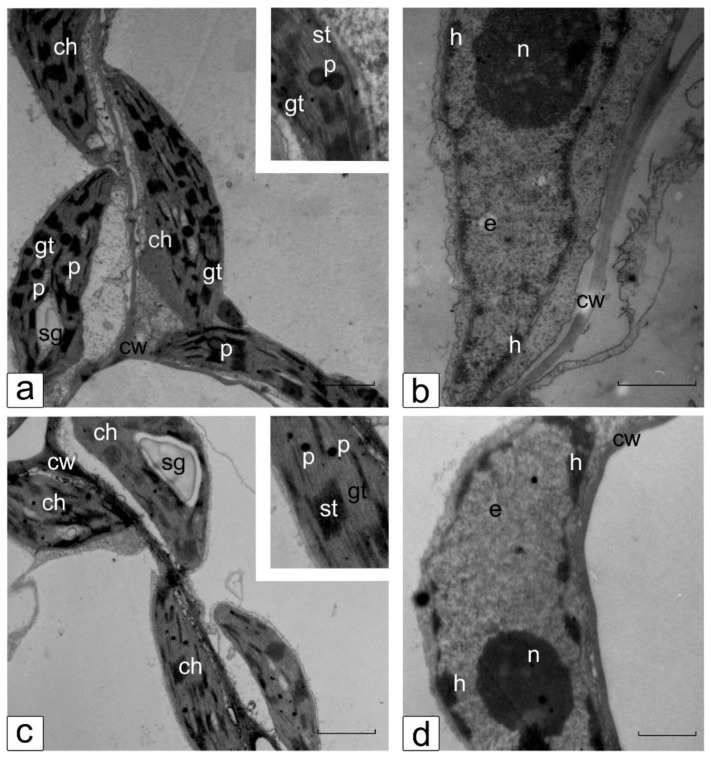
Ultrastructure of chloroplast (**a**,**c**) and nuclear (**b**,**d**) compartments in the cotyledonary SM cells of control (**a**,**b**) and salt-treated (150 mM NaCl) (**c**,**d**) tomato seedlings cv. Rekordsmen. Ch, chloroplast; gt, st, granal and stromal thylakoids, respectively; cw, cell wall; e, euchromatin; h, heterochromatin; n, nucleolus; p, plastoglobuli; sg, starch grain. Scale, 1 µm.

**Figure 9 mps-05-00011-f009:**
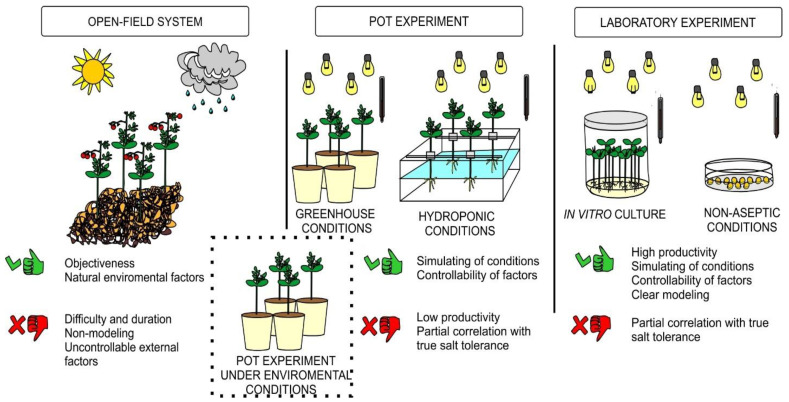
Methods for comparative assessment of tomato salt tolerance.

**Table 1 mps-05-00011-t001:** The effects of different NaCl concentrations in culture medium on the rhizogenesis frequency and time of the beginning root formation from tomato seedlings.

NaCl Concentrations, mM	Rhizogenesis Frequency, %		Time of the Beginning Root Formation, Days	
	Line YaLF	cv. Rekordsmen	Line YaLF	cv. Rekordsmen
0	100	100	4	4
25	100	100	4	4
50	100	100	4	4
75	100	100	4	4
100	100	100	4	4
150	100	100	5	5
200	16,7	100	7	6
250	0	93,3	–	7
300	nd	0	nd	–

Notes: «–», no root formation; nd, not determined.

**Table 2 mps-05-00011-t002:** The effects of different NaCl concentrations in RIM on the RFW and RDW.

NaCl Concentrations, mM	RFW, mg		RDW, mg	
	Line YaLF	cv. Rekordsmen	Line YaLF	cv. Rekordsmen
0	17.68 l	15.97 kl	1.19 l	0.95 k
25	14.08 jk	11.61 ij	0.78 ghijk	0.78 hijk
50	7.44 fgh	8.37 gh	0.67 fghi	0.85 ijk
75	6.54 efgh	8.37 h	0.57 fg	0.91 jk
100	4.78 cdef	5.39 defg	0.47 cdef	0.55 def
150	1.77 ab	3.35 bcd	0.24 b	0.55 ef
200	0.07 a	0.91 ab	0.02a	0.14 ab
250	–	0.70 ab	–	0.08 ab

Notes: «–», no root formation. Means followed by the same letter are not significantly different at α = 0.05 according to the Duncan’s multiple range test (*n* = 30).

## Data Availability

Data sharing is not applicable to this article.

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
