# Peer review of "A Simple and Effective Bioassay Method Suitable to Comparative In Vitro Study of Tomato Salt Tolerance at Early Development Stages"

_mps, 2022, doi:10.3390/mps5010011_

Round 1
Reviewer 1 Report
The manuscript provides an interesting and simple method for evaluating tomato salt tolerance at early developmental stage.
The manuscript is written well enough and can be recommended for publication after correction of some minor remarks:
In the materials and methods section, the information about the tomato genotypes chosen for the study should be expanded ( to describe their characteristics in more detail, taking into account the similarities and differences of the two genotypes).
In the section "discussion", also expand the justification for the choice of these genotypes, because two genotypes is a very small number for the study. The growing regions should also be described in more detail rather than just stating " The soil salinity levels of these regions differ significantly....".
L-93 - give more information about the commercial preparation, (specify the manufacturer, the city of production and the percentage of the main active ingredient). 20% water solution ... - it is not clear what concentration of Cl-containing substance should be used.
L-94 - how many drops of Tween-20 have to be added to a 100ml solution.
Fig. 1 - ... 20% water solution of sodium hypochlorite? - This is unlikely to be possible. Usually commercial solutions contain no more than 5-15% sodium hypochlorite. In order to sterilize plant exhibits, they are diluted 2-3 times to reach a final concentration of 2.5-5%. Please check.
L-109 - specify what membrane was used in filter-sterizetion, not just pore diameter (PES? PVDF? MCE?)
L-129 - specify the model of fluorescent lamp
Author Response
Dear Reviewer,
On behalf of ourselves and the co-authors, we thank you for your appreciation of our manuscript and for your valuable comments and questions. We are confident that your comments and corrections will make better our manuscript.
Remark: In the materials and methods section, the information about the tomato genotypes chosen for the study should be expanded ( to describe their characteristics in more detail, taking into account the similarities and differences of the two genotypes).
In the section "discussion", also expand the justification for the choice of these genotypes, because two genotypes is a very small number for the study. The growing regions should also be described in more detail rather than just stating " The soil salinity levels of these regions differ significantly....".
Response: We detailed information about tomato genotypes (lines 92-93) (F1 Yunior hybrid, and cv. Rekordsmen are recommended for cultivation in the green-house and open-field system, respectively) and soil salinity levels (lines 425-428) (Thus, more than 31% of the soils in the Astrakhan region are characterized by a high salinity (the concentration of sodium and sulfate ions reaches up to 7.1 and 12.5 mM per 100 g of soil, respectively), as well as about 20% of solonetzic soil complexes [54].) in the materials and methods and Results and discussion sections, respectively.
Remark: L-129 - specify the model of fluorescent lamp
Response: The text of the article has been supplemented with the necessary additions of the fluorescent lamp brand (LUMILUX T9 L 32W/840 C G10Q) (line 132).
Remark: L-109 - specify what membrane was used in filter-sterizetion, not just pore diameter (PES? PVDF? MCE?)
Response: We used MCE membrane for sterilization. Text corrected (Line 111).
Remark: L-94 - how many drops of Tween-20 have to be added to a 100ml solution.
Response: Few drops of Tween-20 changed to 5 µl of Tween-20 (Lines 95-96).
Remark: L-93 - give more information about the commercial preparation, (specify the manufacturer, the city of production and the percentage of the main active ingredient). 20% water solution ... - it is not clear what concentration of Cl-containing substance should be used.
Fig. 1 - ... 20% water solution of sodium hypochlorite? - This is unlikely to be possible. Usually commercial solutions contain no more than 5-15% sodium hypochlorite. In order to sterilize plant exhibits, they are diluted 2-3 times to reach a final concentration of 2.5-5%. Please check.
Response: Thank you for your valuable comment. We have made appropriate clarifications in Figure 1., as well as in the Materials and Methods section (Line 95).
Once again, we are so grateful for your review and valuable comments. In addition, we also send you a WORD document with the text, taking into account the comments of another reviewer.
Best regards,
Marat Khaliluev

Reviewer 2 Report
Review report:
The Manuscript entitled “A Simple and Effective Bioassay Method Suitable to Comparative in vitro Study of Tomato Salt Tolerance at Early Development Stages” by Khaliluev et al., reported an in vitro tomato salt stress response evaluation system and showcase its application in differentiating the stress responses of two cultivars at morphological, physiological and cellular levels.
Performing salting stress at rooting stage with shoot apex can avoid the differences caused by seed germination. It is very useful for observing root development related traits, including the number and length of adventitious roots and later roots. The authors also did following up-studies on multiple leaf phenotypes. Overall, it is a comprehensive protocol. Here are my comments:
- Error bars are not included in Figure 2 and Figure 3, while statistic analysis is not shown in Figure 5.
- After series of comprision, the salt stress resistance of line YaLF and cv. Rekordsmen should be summarized.
- In Figure 1, “I step” should be “Step I”.
- The authors should correct the grammatic errors throughout the manuscript.
Author Response
Dear Reviewer,
On behalf of ourselves and the co-authors, we thank you for your appreciation of our manuscript and for your valuable comments. We are confident that your comments and corrections will make our manuscript better.
Remark 1: In Figure 1, “I step” should be “Step I”.
Response 1: We have corrected this inaccuracy. Appropriate changes have been made to Figure 1.
Remark 2. Error bars are not included in Figure 2 and Figure 3, while statistic analysis is not shown in Figure 5.
Response 2: Figures 2 and 3 show the average values, as well as a comparative statistical processing according to the Duncan tests. We also added the results of a comparative assessment of the averages according to the Duncan tests in Fig. 5. (line 308)
Remark 3: After series of comprision, the salt stress resistance of line YaLF and cv. Rekordsmen should be summarized.
Response 3: Thanks for your valuable comment. We added necessary remark.
Thus, based on a comparative assessment, tomato cv. Recordsman characterized by increased resistance to salinity compared to the YaLF line on the organismal (TPR and DRR of seedlings), organ (rhizogenesis frequency; number and length of de novo regenerated roots; root fresh (RFW) and dry (RDW) weights; shoot fresh (SFW) and dry (SDW) weights), tissue (the average cross-sectional area of epidermal and mesophylls cotyledonary cells) and cellular (ultrastructure of chloroplast and nuclear compartments) development levels. (lines 387-393).
Remark 4: The authors should correct the grammatic errors throughout the manuscript.
Response 4: We corrected grammatical errors throughout the manuscript.
Once again, we are so grateful for your review and valuable comments. In addition, we also send you a Word document with the text, taking into account of your comments.
Best regards,
Marat Khaliluev
